# Development and Evaluation/Verification of a Fully Automated Test Platform for the Rapid Detection of *Cyclospora cayetanensis* in Produce Matrices

**DOI:** 10.3390/microorganisms11112805

**Published:** 2023-11-19

**Authors:** Hui Zhu, Beum Jun Kim, Gwendolyn Spizz, Derek Rothrock, Rubina Yasmin, Joseph Arida, John Grocholl, Richard Montagna, Brooke Schwartz, Socrates Trujillo, Sonia Almeria

**Affiliations:** 1Rheonix, Inc., Ithaca, NY 14850, USA; hzhu@rheonix.com (H.Z.); bkim@rheonix.com (B.J.K.); gspizz@rheonix.com (G.S.); drothrock@rheonix.com (D.R.); ryasmin@rheonix.com (R.Y.); rmontagna@rheonix.com (R.M.); bschwartz@rheonix.com (B.S.); 2Division of Virulence Assessment, Office of Applied Research and Safety Assessment, Center for Food Safety and Applied Nutrition, U.S. Food and Drug Administration, Laurel, MD 20708, USA; joseph.arida@fda.hhs.gov (J.A.); john.grocholl@fda.hhs.gov (J.G.); socrates.trujillo@fda.hhs.gov (S.T.); 3Joint Institute for Food Safety and Applied Nutrition, University of Maryland, College Park, MD 20742, USA

**Keywords:** foodborne pathogens, cyclosporiasis, detection, mitochondrial target, rapid test

## Abstract

Cyclosporiasis, caused by the coccidian parasite *Cyclospora cayetanensis*, has emerged as an increasing global public health concern, with the incidence of laboratory-confirmed domestically acquired cases in the US exceeding 10,000 since 2018. A recently published qPCR assay (Mit1C) based on a mitochondrial target gene showed high specificity and good sensitivity for the detection of *C. cayetanensis* in fresh produce. The present study shows the integration and verification of the same mitochondrial target into a fully automated and streamlined platform that performs DNA isolation, PCR, hybridization, results visualization, and reporting of results to simplify and reduce hands-on time for the detection of this parasite. By using the same primer sets for both the target of interest (i.e., Mit1C) and the internal assay control (IAC), we were able to rapidly migrate the previously developed Mit1C qPCR assay into the more streamlined and automated format Rheonix C. cayetanensis^TM^ Assay. Once the best conditions for detection were optimized and the migration to the fully automated format was completed, we compared the performance of the automated platform against the original “bench top” Mit1C qPCR assay. The automated Rheonix C. cayetanensis Assay achieved equivalent performance characteristics as the original assay, including the same performance for both inclusion and exclusion panels, and it was able to detect as low as 5 *C. cayetanensis* oocysts in fresh produce while significantly reducing hands-on time. We expect that the streamlined assay can be used as a tool for outbreak and/or surveillance activities to detect the presence of *C. cayetanensis* in produce samples.

## 1. Introduction

*Cyclospora cayetanensis*, a coccidian parasite that causes protracted gastroenteritis (cyclosporiasis) in humans, has emerged as a global public health concern. Illnesses and outbreaks worldwide have been associated with the consumption of fresh produce such as herbs, berries, and leafy greens [1,2]. Over 10,000 domestically acquired cases of cyclosporiasis have been identified in the US since 2018 [3]. As of 29 August 2023, there have been 1957 cases in 39 states in 2023, with 153 hospitalizations [3]. Recently, a new method for the detection of *C. cayetanensis* in fresh produce was published [4]. The method consists of a qPCR assay (Mit1C qPCR) that uses primers and a probe directed against the mitochondrial cytochrome C oxidase subunit, a conserved region of the mitochondrial genome of *C. cayetanensis*, that varies in other closely related organisms. The use of a mitochondrial gene was originally selected to take advantage of the higher copy number of this gene target as opposed to other molecular targets and the improved specificity of the mitochondrial target region. The real-time PCR assay accurately detects *C. cayetanensis* with an analytical sensitivity of as low as 5 oocysts per sample in produce samples [4]. 

While the sensitive and specific detection of *C. cayetanensis* in produce is possible [4], rapid automated tests for the detection of *C. cayetanensis* contaminating food and environmental samples are still lacking [5]. To further simplify and streamline the detection of *C. cayetanensis*, the Mit1C target was transferred to the Rheonix Encompass Optimum™ workstation. The goal of the present study was to establish and optimize the conditions to detect the mitochondrial target using that workstation and to evaluate and provide a verified and automated method to rapidly and accurately identify *C. cayetanensis* contaminated samples. This method can aid both FDA’s and the food industry’s ability to screen fresh produce for traceback and for risk assessment of agricultural practices and regions. The Encompass platform, coupled with the Rheonix CARD^®^ cartridge technology, has previously been used in food and beverage applications as well as the detection of infectious agents [6,7]. The advantage of the automated platform is that all sample preparation steps (including DNA isolation), PCR amplification, detection of resultant amplicons, and reporting of results can be achieved with minimal hands-on effort. 

By using the same primer sets in the Rheonix C. cayetanensis Assay that were originally employed in the previously described qPCR assay for both the target of interest (i.e., Mit1C) and the internal assay control (IAC), we were able to rapidly migrate the qPCR assay to the more streamlined and automated format. The original Mit1C qPCR assay format employed real-time PCR amplification, while the streamlined format uses an end-point PCR approach. In the end-point PCR approach, the primer sets are biotinylated at their 5′ termini, and after automatic denaturation controlled by the workstation’s software, the amplicons are captured on an integrated DNA array and detected via a colorimetric reaction mediated by streptavidinylated horse radish peroxidase [8,9]. The availability of a more streamlined assay and accessible format can help reduce the time, complexity, and labor associated with screening food and environmental samples for traceback and risk assessment by food producers and public health agencies. 

The fully automated assay platform utilizes the Rheonix Optimum workstation (Figure 1), into which the disposable microfluidic CARD cartridges (Figure 2) are inserted. Each CARD cartridge processes four independent specimens, and the workstation can process six separate CARD cartridges, bringing the throughput to 24 individual samples per run. 

Once the migration to the fully automated format was completed and the best conditions set up, we compared the performance of the automated platform against the original “bench top” qPCR assay and demonstrated that the analytical sensitivity was equivalent to the original assay and had the same performance for both inclusion and exclusion panels. 

## 2. Materials and Methods

### 2.1. Cyclospora cayetanensis Oocysts

Purified *C. cayetanensis* oocysts from a patient diagnosed with cyclosporiasis in Guatemala were kindly supplied by the Center for Disease Control and Prevention (CDC). The use of the purified oocysts from clinical specimens was approved by the Institutional Review Board of the FDA (Protocol 15-039F and RIHSC-ID#10-095F).

The purified *C. cayetanensis* oocysts, which had been stored in 0.25% potassium dichromate, were washed three times in 0.85% sodium chloride and concentrated by centrifugation [10]. Six replicates of the concentrated oocyst preparation were enumerated using a hemocytometer. Dilutions of estimated amounts of oocysts were then prepared for inoculation as previously described [10].

### 2.2. Processing of Fresh Produce Samples in the Development and Evaluation of the Fully Automated Rheonix C. cayetanensis Assay

Samples consisted of 25 g of commercial fresh produce and prepared dishes with fresh produce, except for fresh berry test samples, which consisted of 50 g. Fresh produce samples were inoculated with the dilutions of estimated numbers of purified oocysts. Inoculation and washing of produce samples followed the procedures previously described [11,12]. Unseeded samples were processed as negative controls at the same time as seeded samples.

The development, evaluation, and verification of the fully automated Rheonix C. cayetanensis Assay in fresh produce was accomplished in two phases. In the first phase of the work (phase I), we demonstrated the detection of Mit1C on the Encompass Optimum workstation using DNA that was manually isolated and purified. In the second phase of the work (phase II), we developed and verified a streamlined assay format in which the entire assay, including sample preparation, analysis, and readout, was performed in a fully integrated and automated manner. For phase I, after seeding of produce and washing the oocysts from the fresh produce samples, genomic DNA extraction was performed. Genomic DNA was isolated following the FDA BAM 19b method [11]. Briefly, DNA from the produce samples was isolated using the FASTDNA^TM^ SPIN Kit for Soil in conjunction with the FastPrep-24 Instrument (MP Biomedicals, Santa Ana, CA, USA), following the manufacturer’s instructions with slight modifications as previously described [11,12]. The total elution was 75 µL of DNA, and 2 µL of DNA/20 µL reaction was evaluated by the real-time Mit1C PCR assay of the *C. cayetanensis* mitochondrial gene target [4]. Each sample was run in triplicate, and samples were considered positive for the presence of *C. cayetanensis* when at least one replicate produced a positive result in the real-time PCR [4,11]. DNA from seeded samples with different numbers of *C. cayetanensis* oocysts previously found positive by Mit1C qPCR and unseeded control produce samples negative for the presence of parasite DNA were added directly into the CARDs (Figure 2) and processed automatically for PCR amplification and detection in the Rheonix C. cayetanensis Assay.

For Phase II development and verification, once produce samples were washed and oocysts concentrated [11,12], only the first steps of the genomic DNA extraction were manually performed to break the oocyst walls and release their genomic material. Briefly, the washed pellet for each produce sample was put into a Lysing Matrix E tube (supplied with the FastDNA Spin Kit) containing beads and 122 µL of MT buffer and 978 µL of Sodium Phosphate Buffer (all supplied with the FastDNA Spin Kit) were added. After two cycles in the FastPrep-24 bead beater (6.5 m/s, approximately 4000 rpm, for 60 s, twice), tubes were removed and centrifuged at 14,000× *g* for 15 min. The supernatant (approximately 1 mL) was transferred to a clean Eppendorf tube and used in the integrated fully automated Rheonix C. cayetanensis Assay, which included DNA isolation, PCR amplification, and detection, with a loading volume of 200 µL/sample. If the bead-beaten supernatant was not used immediately, it was kept at −20 °C until analysis. 

### 2.3. Inclusivity and Exclusivity Panel Analysis for Specificity and Sensitivity of the Rheonix C. cayetanensis Assay

DNA from 12 clinical samples generously donated by CDC, containing *C. cayetanensis* oocysts and found to be positive by qPCR Mit1C [4], were analyzed as an inclusivity panel in the Rheonix C. cayetanensis Assay for sensitivity purposes. The clinical samples were stored in a fixative or in a Cary-Blair nonnutritive medium. DNA was extracted from 200 µL of homogenized clinical sample using the FASTDNA^TM^ SPIN Kit for Soil in conjunction with the FastPrep-24 Instrument as indicated for produce samples [4]. 

Eleven parasites closely related to *C. cayetanensis* were selected for the exclusivity panel for purposes of specificity. *Eimeria tenella*, *Eimeria maxima*, and *Eimeria acervulina* oocysts were kindly gifted by USDA ARS (Beltsville, MD, USA), and DNA was extracted from those isolates following the same procedure as for clinical samples. All additional purified DNA samples for use in the exclusivity panel were purchased from ATCC. Organisms tested included *Blastocystis hominis*, *Giardia intestinalis*, *Entamoeba histolytica*, *Cryptosporidium parvum*, *Plasmodium falciparum*, *Toxoplasma gondii*, *Trypanosoma cruzi*, and *Neospora caninum*. Those parasite species had been previously tested as part of the inclusivity and exclusivity panels for Mit1C qPCR [4]. Two microliters of DNA of each sample in the inclusivity and exclusivity panels were used as a template for Encompass assays as had been used for qPCR Mit1C. 

### 2.4. General Procedure for Performance of the Rheonix C. cayetanensis Assay 

The primer and probe sets originally reported [4] for the amplification and detection of the Mit1C and internal assay control (IAC) in the real-time qPCR assay (Table 1) were used in the automated version of the assay with the exception that the 5′ termini of the reverse primers were biotinylated for use in the end-point PCR reaction performed in the automated assay. The fully automated version of the assay also used the same probe sequences that were employed in the “bench top” qPCR assay, with the exception that instead of using fluorescently tagged probes, the probes were aminoterminated at their 5′ termini to allow for covalent linkage to the membrane filter via 1-ethyl-3-(3-dimethylaminopropyl) carbodiimide (EDAC) chemistry [10].

The Rheonix Encompass Optimum workstation can perform the fully automated assay with minimal intervention from the user. The workstation’s software controls all assay functions, including the control of the pneumatic signals required to actuate the various pumps and valves contained within the CARD cartridge, timing, and temperature control of the various steps, including thermocycling conditions. The software also controls the onboard CMOS (complementary metal-oxide semiconductor) camera used to acquire the hybridization signals, which are interpreted by the workstation’s software and automatically reported either via the touchscreen user interface, a connected printer, or connection to a laboratory’s LIS system. Briefly described, the workstation’s robotic arm picked up individual pipette tips to aspirate and transport the individual specimens from the sample tubes to the appropriate reservoirs on the CARD cartridge. Two hundred microliters of the specimens were introduced by the robotic arm to the CARD cartridge. Then, as needed, the robotic arm also picked up pipette tips to deliver the required reagents contained in the disposable reagent pack to the CARD cartridges. After the specimens were delivered to the CARD cartridges, 190 µL of lysis buffer (2.5 M of guanidine hydrochloride, 50 mM of MOPS, 10 mM of EDTA, and 1% of Triton X-100, pH 7.0) were added to the samples and incubated for 10 min at 60 °C, followed by adding 20 µL of Proteinase K (Sigma Aldrich, St. Louis, MO, USA, Cat# P4850) and lysis buffer mixture at 3:2 ratio and continuing to incubate for 5 min at 60 °C. Then, 10 µL of paramagnetic beads (AMSBIO, Cambridge, MA, USA, Cat# MD120006) and 230 µL of isopropyl alcohol were added and incubated for 10 min at 35 °C. The workstation then activated magnets that are located under the appropriate reservoirs of the CARD cartridges to capture the magnetic bead-nucleic acid complexes. The beads were washed with Wash Buffer 1 (40% Isopropanol, 0.8 M guanidine hydrochloride, 15 mM MOPS, pH 6.7) and Wash Buffer 2 (70% ethanol, 150 mM NaCl, 15 mM MOPS, pH 6.7) two times each, drying for 8 min, and then the nucleic acids were eluted from the complexes with ~30 µL of water and 6 µL was delivered to the PCR reaction chambers followed by 9 µL of PCR master mix, which was mimicked in the originally reported real-time PCR reaction mix (Table 2), as shown in Table 3. After the thermocycling reactions were completed, 7 µL of the PCR amplicons and ~93 µL of SS Buffer (1X SSPE, 0.1% SDS, pH 8.0) were then delivered to the DNA arrays via pneumatic pumping action, and the amplicons were allowed to hybridize to the DNA probes specific for the targets of interest for 12 min at 57 °C or other designated temperature. After washing with 100 µL of SS Buffer, 110 µL of streptavidinylated horseradish peroxidase (HRP—Thermo Fisher Scientific, Waltham, MA, USA, Cat# 21126) were added and incubated for 5 min at 35 °C. After washing with 100 µL of SS Buffer four times, 170 µL of 3,3′, 5, 5′-Tetramethylbenzidine (TMB, SurModics, Inc., Eden Prairie, MN, USA, Cat# TMBM) were added and incubated for 4 min at 35 °C to facilitate the oxidation of the TMB substrate by HRP. After the completion of the HRP-TMB reaction, 100 µL of 75% glycerol was added to prime the filter for image analysis. Then, the CMOS camera attached to the robotic arm imaged the hybridization spots, and using the established algorithm, the intensity of each of the hybridization spots was classified as either positive, negative, or indeterminate for the targets of interest.

### 2.5. Comparison of the Performance of the Real-Time Mit1C PCR versus the Rheonix C. cayetanensis Assay

When the assay was performed on the fully automated Rheonix C. cayetanensis Assay, the final volume of the PCR reaction was adjusted to 15 µL, but the concentrations of all assay components remained the same. Two master mixes (a) PerfeCTa multiplex qPCR Toughmix Low ROX and (b) PCRBio master mix and dUTP were analyzed and compared in the Rheonix C. cayetanensis Assay (Table 3).

## 3. Results

### 3.1. Selection of Probe Concentrations and Hybridization Temperatures for the Rheonix C. cayetanensis Assay 

Amplicons for the Mit1C and IAC targets were generated via bench top PCR and were analyzed on 2% agarose gel to confirm the amplicon sizes. As expected, the Mit1C amplicons and IAC amplicons displayed bands at 205 base pairs and 145 base pairs on gel electrophoresis, respectively. Then, those amplicons were allowed to react with the immobilized target-specific probes on the DNA arrays. Simultaneously, different probe concentrations and hybridization temperatures were evaluated. Empirically, we demonstrated that a probe concentration of 1–80 µM is a good range for target detection in this system. Tm for the Mit1C probe and IAC probe were 64.6 °C and 68.3 °C, respectively, and we typically set the hybridization temperature 10 °C below the Tm. To optimize these parameters, various concentrations ranging from 1.25 µM to 80 µM of both probes on a single DNA array were tested (Figure 3) at different hybridization temperatures ranging from 48 °C to 60 °C (Figure 4), which could be controlled independently on each of the six stations on the deck of the workstation. The DNA array results showed both Mit1C and IAC targets were successfully detected at their corresponding spots (Figure 4) with different signal intensities (Figure 5).

Probe concentrations were then finalized at 80 µM and 20 µM for Mit1C and IAC, respectively, and the hybridization temperature was set at 57 °C for further studies. 

### 3.2. Assessing Assay Sensitivity and Potential Matrix Inhibitors in the Rheonix C. cayetanensis Assay 

Isolated DNA (approximately 80 µL) from romaine lettuce samples seeded with 200 *C. cayetanensis* oocysts was used to assess assay sensitivity and potential matrix inhibition to the PCR reaction, which is not uncommon in other *C. cayetanensis* PCR-based assays. Different amounts of DNA eluent (i.e., 1.5, 3.0, 4.5, and 6 µL), along with other PCR components in Table 3, were then introduced into a 15 µL of PCR reaction on the CARD cartridges, run in triplicate (Figure 6), and automatically processed via PCR amplification and detection. Using the thermocycling conditions described in Table 4, the fully automated platform was able to successfully detect the amplicons for both Mit1C and the IAC targets at all the DNA input amounts, indicating no matrix inhibition was observed with up to 6 µL of DNA input from romaine lettuce seeded with 200 *C. cayetanensis* oocysts (Figure 6), and as low as approximately 3.75 oocysts per PCR reaction was detected. For further analysis, 6 µL of DNA in a 15 µL reaction were used.

### 3.3. Elimination of Potential Carry-Over Contaminations and Comparison of Master Mixes in the Rheonix C. cayetanensis Assay

To eliminate potential amplicon carry-over contamination and increase the stringency of the Rheonix C. cayetanensis Assay, UNG (Uracyl N-Glycosylase) and dUTP, which are routinely applied to the PCR master mixes of other Rheonix assays, were also included in the PCR reactions. The PCRBio master mix, with the addition of UNG and dUTP at a final concentration of 0.01 unit/µL and 0.5 mM, respectively, were tested in comparison to the PerfeCTa master mix, which was initially adapted from the bench top qPCR test, with the automated process via PCR amplification and detection in duplicate samples. The results showed the PCRBio master mix with UNG and dUTP had very similar assay sensitivity as compared to the PerfeCTa master mix (Table 5).

Further determination of the limit of detection (LoD) in copies/reaction for both the PCRBio master mix with UNG and dUTP and PerfeCTa master mix were evaluated by 2-fold dilutions of Mit1AA gBlock (a 245 bp synthetic DNA fragment [4]) positive control starting at 6.25 copies/reaction. Fractional levels (25–75%) were detected at 3.12 to 1.56 copies/reaction using the PerfeCTa master mix, while PCRBio was capable of detecting fractional levels of 0.78 copies/reaction. The results reinforced that the PCRBio master mix with UNG and dUTP had a similar assay sensitivity and LoD compared to the PerfeCTa master mix (Table 6).

For PCRBio master mix with UNG and dUTP LoD95%, the concentration of 6.25 copies/reaction was run in many replicates (21 replicates) with positive results in all replicates, demonstrating the capacity of detecting at least 6.25 copies/reaction of *C. cayetanensis* in a reaction with a high probability of repeatability. The LOD95% was also computed via a web service (https://quodata.de/content/validation-qualitative-pcr-methods-single-laboratory)(accessed on 2 October 2023). The calculated PerfeCTa LOD95% was 4.140 with a 95% confidence interval of [1.898, 9.011], while PCRBio LOD95% was 6.670 with a 95% confidence interval of [3.422, 12.500]. Based on the early study [4], the LOD95% of Mit1C qPCR was computed to be 3.171 copies with a 95% confidence interval of [2.002, 5.037] using synthetic Mit1AA gBlock standards DNA.

### 3.4. Inclusivity/Exclusivity Panel Results

All clinical samples included in the inclusivity panel (n = 12), which were positive by qPCR, produced positive results in the Rheonix C. cayetanensis Assay, irrespective of their geographical location. Moreover, no cross-reactions were observed when analyzing DNA recovered from closely related parasites of the Apicomplexa phylum (n = 11) via the Rheonix C. cayetanensis Assay included in the exclusivity panel. 

### 3.5. Verification of Rheonix Automated PCR Assay in Produce Samples Using DNA Manually Extracted (Phase I)

In phase I, fresh produce samples DNA (a total of 21 samples) which had been previously analyzed by Mit1C qPCR, including lettuce (mixed lettuce, romaine lettuce, and/or baby romaine lettuce), carrots, basil, berries (blackberries, blueberries, and/or raspberries), and commercially prepared pico de gallo unseeded and seeded with 200, 50, 20, 10, or 5 *C. cayetanensis* oocysts were analyzed in the Rheonix C. cayetanensis Assay (Table 7). Two microliters of DNA were used per reaction for the high-seeding samples (200, 50, and 20 oocysts), and 2 µL and 8 µL of DNA were used for low-seeded samples (5 or 10 oocysts) as a template for the Rheonix assays. All unseeded samples (Table 7) were found to be negative by the Rheonix C. cayetanensis Assay. All previously positive samples of fresh produce by qPCR were also found positive, with the exception of two low-seeded samples, one seeded pico de gallo sample with 5 oocysts and one carrot sample seeded with 10 oocysts, which were positive when using 8 µL of DNA in the Rheonix system. The amount of DNA (8 µL of DNA in a 20 µL reaction) was equivalent to 6 µL DNA in a 15 µL reaction selected previously in the set-up of the Rheonix Assay. The results showed that the Rheonix C. cayetanensis Assay had comparable detection results as the qPCR, and it was able to detect as low as five *C. cayetanensis* oocysts in different fresh produce food types analyzed when DNA manually extracted was directly added to the CARDs in the same amounts as in the qPCR.

### 3.6. Phase II: Development of the Fully Automated PCR Reaction

Once the conditions for amplification and detection of the targets of interest were established for the fully automated Rheonix C. cayetanensis Assay, we then turned our attention to adding the DNA isolation and purification to the fully automated assay employing bead-beaten produce samples seeded with *C. cayetanensis* oocysts at various concentrations to establish the fully integrated automated assay. 

Romaine lettuce samples seeded with 50 oocysts, pretreated with bead-beating, and resulting in ~1 mL of volume, were serially diluted down to 1.56 oocyst-seeded-equivalent with negative samples (negative unseeded romaine lettuce beat-beaten samples) and 200 µL of the samples were loaded on the CARD. The fully automated Rheonix C. cayetanensis Assay was able to fully integrate DNA purification, PCR amplification, and amplicon detection under the control of the workstation’s software, resulting in easily detectable hybridization signals. As low as 3.13-oocyst-seeded-equivalent romaine lettuce sample was detected in the fully integrated automated process (Figure 7), confirming the initial results using manually extracted DNA from samples seeded with as low as 5 oocysts.

To have IAC serving not only as internal amplification control but also as true process control, the fully integrated and automated Rheonix C. cayetanensis Assay process was finalized by spiking 400 µL of IAC gBlock, a 2000 bp synthetic DNA fragment based on IAC Ultramer ^TM^ sequence [4], at 500 cp/µL into lysis buffer in the reagent kit by the robotic arm of the workstation instead of adding it into the PCR master mix, such that the IAC would undergo the entire purification, amplification, and detection process as the Mit1C target in the samples. In theory, it would result in ~1000 copies of IAC per PCR reaction, assuming 100% recovery during purification. 

The resulting fully integrated and automated Rheonix C. cayetanensis Assay for 24 samples took approximately 4 h to complete without any hands-on efforts after the initial loading of the prepared samples, reagents, and consumables onto the workstation (15 min hands-on time). 

### 3.7. Verification of Final Assay Format for Fully Automated PCR Assay in Produce Samples (Phase II) 

To verify the performance characteristics of the fully automated Rheonix C. cayetanensis Assay, bead-beaten produce samples seeded with *C. cayetanensis* oocysts at various concentrations were used in the fully integrated, automated assay. Produce samples included cilantro, romaine lettuce, and berries in unseeded samples and samples seeded with high levels (100 oocysts) and low levels of *C. cayetanensis* oocysts (5 oocysts) following the Guidelines for the Validation of Microbiological Methods for the FDA Foods Program, 3rd edition [13] (Table 8). In the real-time qPCR assay, the whole supernatant of a bead-beaten sample (approximately 1 mL) is processed to isolate DNA manually. Therefore, to compare the detection of *C. cayetanensis* in the supernatant of samples in the Rheonix C. cayetanensis Assay in low-seeded samples, which are expected to have fractional rates of positivity (50% ± 25%), the supernatant of each bead-beaten sample was analyzed in quadruplicate (250 µL each replicate) and samples were considered positive for the presence of *C. cayetanensis* when at least one replicate produced a positive result in the Rheonix C. cayetanensis Assay.

In the Rheonix C. cayetanensis Assay, all unseeded samples were negative, and all samples seeded with high levels of oocysts were positive. The positive rates in romaine lettuce and cilantro samples seeded with 5 oocysts were 80% for both commodities. No inhibition was observed when unseeded or any seeded samples of cilantro and romaine lettuce were analyzed. Rates of positivity in the fresh produce of these commodities by Mit1C qPCR in samples seeded with 5 oocysts were previously reported [4], 75% in cilantro and 50% in romaine lettuce. Therefore, increased rates of positivity in the supernatant of bead-beaten samples of cilantro and romaine lettuce seeded with 5 oocysts were observed in the Rheonix C. cayetanensis Assay.

Inhibited reactions were observed when bead-beaten supernatants from raspberry samples were analyzed in the Rheonix C. cayetanensis Assay. When samples of bead-beaten supernatants of raspberries were diluted with water (Table 9), assay sensitivity was improved.

Inhibition was not observed when unseeded samples from blueberries (n = 3) and blackberries (n = 3) were analyzed. Bead-beaten supernatants of seeded blueberries with 5 oocysts (n = 5) were processed, and, similarly, no inhibition was observed. The positive rate in blueberries seeded with 5 oocysts was 40%, with two positive samples showing two positive replicates each.

## 4. Discussion

The continual development of new or improved methods for the detection of foodborne pathogens in produce and in the produce production and processing environments is a priority for the FDA and public health agencies [4]. There are sensitive and specific methods for the detection of *C. cayetanensis* in produce [4,5,10]. However, to our knowledge, there were not any rapid automated tests for the detection of *C. cayetanensis,* contaminating fresh produce and environmental samples [5].

Although there are some automated systems that can detect multiple gastro-intestinal (GI) pathogens, including *C. cayetanensis* (reviewed by [14]), those have been used for detection in clinical samples, which usually have high numbers of *C. cayetanensis* oocysts. On the other hand, fresh produce detection has different challenges, and only a small number of oocysts are present in contaminated produce. Other challenges in the analysis of fresh produce include their short shelf life, the long period for *Cyclospora* infection to show any symptoms (over one week), and the complexity of tracing the source of produce contamination [12]. In addition, *C. cayetanensis* cannot be propagated in vivo or in vitro to confirm its presence in foods or environmental samples. Therefore, molecular methods are crucial for *C. cayetanensis* detection in produce because of their high sensitivity and specificity and rapid analysis for detection. In the present study, we developed and evaluated a fully automated test platform for the rapid qualitative detection of *C. cayetanensis* in produce matrices. Although not a real-time PCR assay, the present Rheonix C. cayetanensis Assay method is a non-subjective, high-throughput tool for detection in fresh produce, making it useful for routine surveillance of fresh produce for *C. cayetanensis* oocysts. An additional advantage of the new automated Rheonix C. cayetanensis Assay for detection in fresh produce is the significantly reduced hands-on time in the processing of samples and reporting of results (approximately 4 h of technician hands-on time saved for 24 samples). Finally, due to the workstation’s ability to automatically perform all preparative, analytical, and readout functions, the expense of some pieces of equipment could also be eliminated in testing laboratories, thus further reducing overall costs and increasing available laboratory space to those labs performing *C. cayetanensis* testing [7,8].

The newly developed Rheonix C. cayetanensis Assay showed comparable specificity and comparable sensitivity, as shown in the exclusivity and inclusivity panels with no cross-reactions observed with any of the closely related parasites analyzed, and it was able to detect as few as 5 oocysts in fresh produce of different commodities, as has been previously reported using real-time qPCR for this parasite [4,10,12]. The performance of the Rheonix C. cayetanensis Assay was initially assessed by analysis of serial dilutions of the *C. cayetanensis* positive gBlock, comparing two different master mixes, PerfeCTa Quantabio and PCRBio master mix with UNG and dUTP and both showed very similar assay sensitivity. Both master mixes also showed similar LoD with low copy numbers being able to be detected (The average calculated LOD_95%_ was 4.140 for PerfeCTa and 6.670 for PCRBio). The PCRBio master mix with UNG and dUTP was selected to eliminate potential amplicon carry-over contamination and to increase the stringency of the Rheonix C. cayetanensis Assay. When manually extracted DNA was used side-by-side in the comparison with the real-time PCR Mit1C method in fresh produce matrices, the results were comparable and able to detect as few as five seeded *C. cayetanensis* oocysts in different fresh produce matrices. All produce samples seeded with 100 oocysts were positive, and all unseeded samples from each fresh produce were negative. No matrix inhibition was observed with up to 6 µL of DNA input in the Rheonix C. cayetanensis Assay based on the IAC signal of any sample processed. The DNA amount needed to be able to consistently detect low levels of oocysts was established as 8 µL/20 µL reactions that were proportional to the final amount established in the final automated assay (6 µL/15 µL reaction). No matrix inhibition was observed at those conditions in fresh produce.

Once we demonstrated the level of detection of *C. cayetanensis* Mit1C on the Encompass Optimum workstation using DNA that was manually isolated and purified (phase I), for phase II, the entire assay, including sample preparation (DNA extraction), was performed in a fully integrated and automated manner. A challenge of nucleic acid isolation from *C. cayetanensis* oocysts, as with other parasites, from helminthic eggs to protozoan oocysts, is their thick and tough protective exterior wall [15,16]. Without breaking the exterior walls, conventional DNA isolation systems provide poor performance in extracting parasite DNA from different matrices [15], and that was the case when a method using DNA isolation based on vortexing of samples was tested with *C. cayetanensis*, which could be explained as not being enough to break the oocyst wall and extract DNA from most oocysts of *C. cayetanensis* [17]. Bead-beating is the most consistently used method for the break-up of walls in the DNA isolation of *C. cayetanensis* oocysts (reviewed by [17]). Therefore, bead-beating supernatants from washed fresh produce pellets were selected for the full automated detection of *C. cayetanensis* in the Rheonix Assay (phase II).

Based on the Guidelines for the Validation of Microbiological Methods of the FDA Foods Program [13], unseeded samples and two levels of seeding samples (high and low) are needed for validation studies. In real-time PCR assays, unseeded samples are expected to always be negative, and high-seeded samples (100 oocysts in this study) are expected to always be positive, while the fractional level of positivity (50% ± 25%) are expected for low-seeded samples (5 oocysts in this study). In addition, samples are considered positive for the presence of *C. cayetanensis* when at least one replicate of three produces a positive result in the real-time PCR [11]. In our study, to indirectly compare detection rates, the total volume of bead-beaten supernatant of each sample was analyzed in quadruplicate (250 µL in each replicate), and samples were considered positive for the presence of *C. cayetanensis* when at least one replicate produced a positive result in the Rheonix C. cayetanensis Assay. When the full automated assay was performed on those samples derived from wash-produce supernatants after bead-beating in high-risk produce such as herbs, leafy greens, and berries, results confirmed the excellent rate of positivity observed in samples seeded with low levels of oocysts (5 oocysts), with high rates of positivity (80% in cilantro and romaine lettuce). Some inhibition issues were observed in the processing of bead-beaten raspberry samples. The detection of *C. cayetanensis* oocysts in fresh produce in general and in berries is challenging since the PCR assay must be performed directly on nucleic acids extracted from food samples that may contain inhibitors and high levels of background DNA. Some polyphenol inhibitors are found in berries, including flavonoids (anthocyanins, flavonols, flavones, flavanols, flavanones, and isoflavonoids), stilbenes, tannins, and phenolic acids [18]. However, no inhibition was observed in the processing of other fresh berries (blackberries and blueberries). Thus, the reason for this inhibition only in raspberries is unclear and could be due to the presence of different components, including flavonols and flavons, anthocyanins, phenolic acids, and/or hydrolyzable tannins in each berry type [19]. If inhibition occurs when processing raspberries, a dilution of the samples to 1/4 or 1/10 will be needed before samples are included in the Rheonix C. cayetanensis Assay.

As a limitation of this method, as with any molecular method for detection based on DNA, the viability of oocysts in fresh produce samples cannot be verified. Molecular methods are useful to determine levels of contamination in produce, but they detect live and infectious, live and non-infectious, and dead oocysts. Another limitation of the study is that in spiking experiments, there is invariably some inconsistency in the exact number of oocysts seeded per sample due to pipetting variability, particularly with low oocyst counts [12].

## 5. Conclusions

In conclusion, the new automated Rheonix C. cayetanensis Assay showed consistent and high detection rates for the detection of *C. cayetanensis* in samples of high-risk fresh produce matrices such as herbs, leafy greens and berries with low levels of oocysts. This streamlined assay can be used as a tool for outbreak and/or surveillance activities to detect the presence of *C. cayetanensis* in produce samples.

## Figures and Tables

**Figure 1 microorganisms-11-02805-f001:**
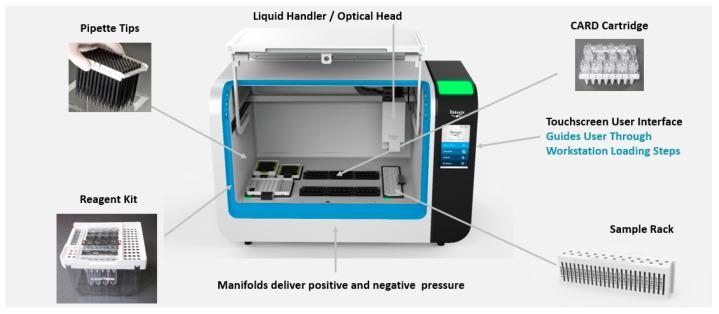
Rheonix Encompass Optimum workstation. Location of all disposables and sample rack as introduced onto the deck of the workstation. The liquid handler and optical head are shown on the robotic arm. All assay processes (DNA isolation, PCR, hybridization, results visualization, and reporting), as well as signal acquisition and reporting of results, are under the control of the workstation’s software. The touchscreen user interface is used by the operator to access assay-specific information, initiate the run, and view results.

**Figure 2 microorganisms-11-02805-f002:**
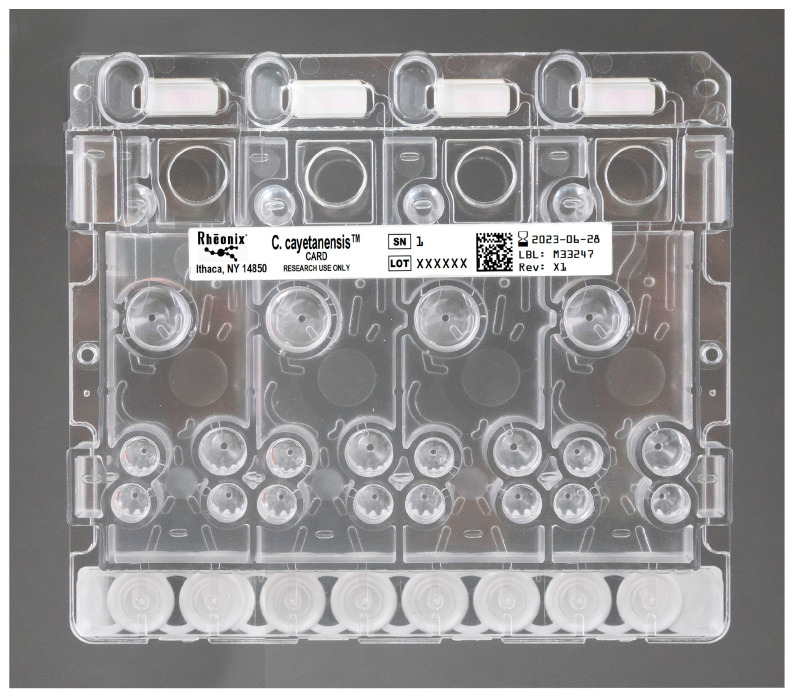
Rheonix CARD cartridge. Rheonix CARD cartridge capable of processing four separate specimens. The robotic arm of the Rheonix Optimum workstation picks up pipette tips from within the workstation and uses them to deliver specimens and the various reagents required to perform all the reactions, including PCR, into the appropriate reservoirs. The PCR reaction tubes are shown at the bottom of the CARD cartridge, and the integrated DNA array containing probes against the *C. cayetanensis* Mit1C and internal amplification control (IAC) amplicons are shown across the top of the CARD cartridge. All waste generated by the reactions is retained within the cartridge, allowing the entire CARD cartridge to be disposed of as biohazardous waste.

**Figure 3 microorganisms-11-02805-f003:**
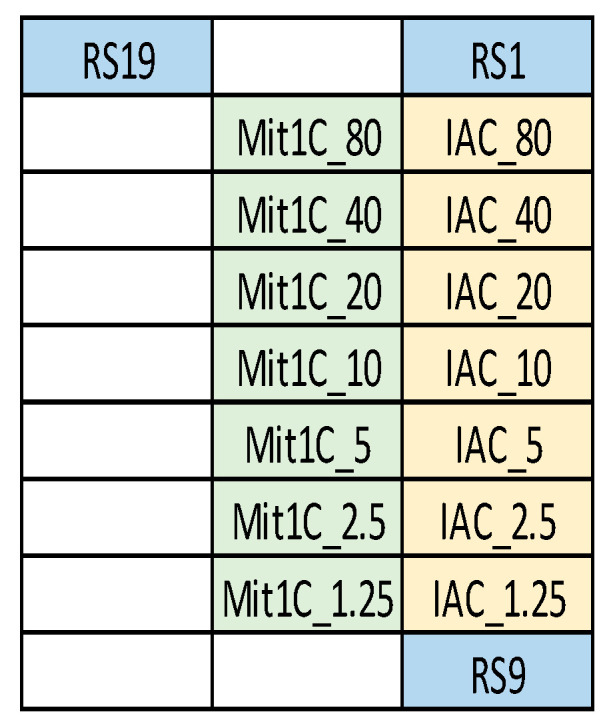
Configuration of DNA arrays used in CARD cartridge. Probes against Mit1C and IAC were spotted on the DNA array at concentrations ranging from 1.25 µM to 80 µM, as noted. Spots designated as RS1, RS9, and RS19 represent an 18mer biotinylated oligo dT, which is used as one of the controls to demonstrate the horseradish peroxidase, TMB reaction performed correctly. Regardless of the presence of any of the target hybridization spots, the RS spots must be present for the assay to be considered valid.

**Figure 4 microorganisms-11-02805-f004:**
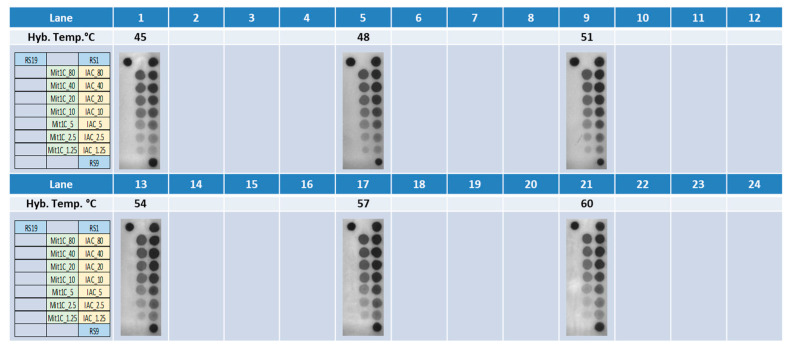
DNA Array results. The amplicons evaluated by 2% gel electrophoresis were allowed to hybridize to DNA probes specific for Mit1C and IAC targets at different concentrations and at various hybridization temperatures.

**Figure 5 microorganisms-11-02805-f005:**
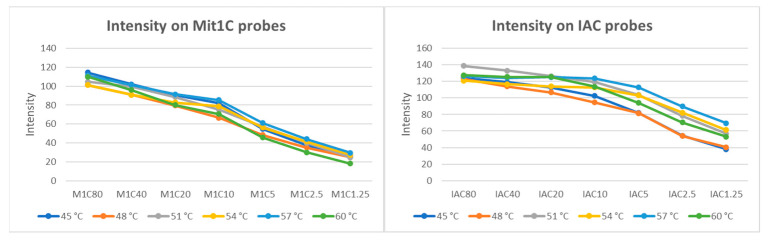
Signal intensity of hybridization. Signal intensity of each probe concentration at different hybridization temperatures. (**Left**) shows Mit1C intensity, and (**right**) shows the IAC intensity.

**Figure 6 microorganisms-11-02805-f006:**
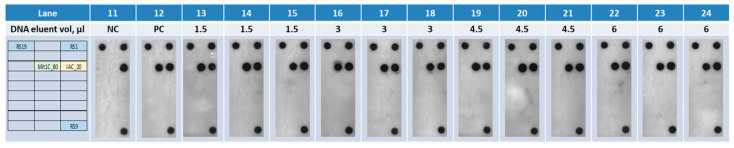
Amplification and detection of DNA targets (obtained by manual methods) on the fully automated assay platform using different amounts of DNA in triplicate.

**Figure 7 microorganisms-11-02805-f007:**
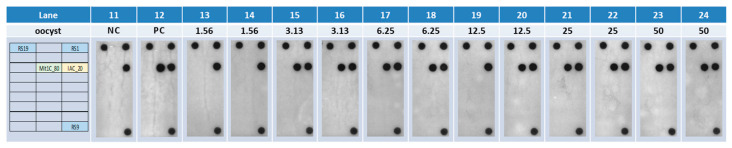
*Cyclospora cayetanensis* oocyst-seeded romaine lettuce samples (50 oocysts) were serially diluted (duplicates), purified, amplified, and detected in a fully integrated automated assay, as low as 3.13-oocysts-seeded equivalent sample was detected.

**Table 1 microorganisms-11-02805-t001:** PCR Primers and Probes used in the *C. cayetanensis* Mit1C-IAC Assay [4].

	Mit1C	IAC
Forward Primer	5′-TCTATTTTCACCATTCTTGCTCAC-3′	5′-CTAACCTTCGTGATGAGCAATCG-3′
Reverse Primer	5′-TGGACTTACTAGGGTGGAGTCT-3′	5′-GATCAGCTACGTGAGGTCCTAC-3′
Amplicon Size	205 bp	145 bp
Probe	5′-AGGAGATAGAATGCTGGTGTATGCACC-3′	5′-AGCTAGTCGATGCACTCCAGTCCTCCT-3′

**Table 2 microorganisms-11-02805-t002:** PCR Reaction Mix for real-time detection of Mit1C and IAC.

Component	Final Concentration
Nuclease-free water	Variable
PerfeCTa Multiplex qPCR ToughMix Low ROX (5×)	1×
Mit1C primer (Forward)	400 nM
Mit1C primer (Reverse)	400 nM
Mit1C Probe	250 nM
IAC DNA (500 copies/µL)	2 µL
IAC primer (Forward)	400 nM
IAC primer (Reverse)	400 nM
IAC Probe	250 nM
Template	Variable
Total Volume	20 µL

PerfeCTa multiplex qPCR Toughmix Low ROX, Quantabio LLC., Beverly, MA, USA, Cat# 95149-250.

**Table 3 microorganisms-11-02805-t003:** PCR Reaction Mix for fully automated detection of Mit1C and IAC using PerfeCTa or PCRBio master mix.

Component *	Final Concentration
Nuclease-free water	Variable
PerfeCTa Multiplex 5× or PCRBio 4×/dUTP/UNG	1×
Mit1C primer (Forward)	400 nM
Mit1C primer (Reverse)	400 nM
IAC DNA (500 copies/µL)	1.5 µL
IAC primer (Forward)	400 nM
IAC primer (Reverse)	400 nM
Template	Variable
Total Volume	15 µL

* The probes in the Rheonix Assay are spotted on the filter. (a) PerfeCTa multiplex qPCR Toughmix Low ROX, Quantabio LLC., Beverly, MA, USA, Cat# 95149-250 and (b) PCRBio master mix (PCR Biosystems, London, UK, #PB25.53-P4) with UNG (ArcticZymes, Tromso, Norway, Cat#75011) and dUTP (Promega, Madison, WI, USA, Cat# U1191).

**Table 4 microorganisms-11-02805-t004:** Thermocycling conditions used in the Encompass Optimum workstation.

PCR Step	Temperature/Time/Number of Cycles
Initial denaturation	96 °C/3 min/1 cycle
Heat denaturation	96 °C/15 s/40 cycles
Annealing/elongation	62 °C/1 min/40 cycles

**Table 5 microorganisms-11-02805-t005:** Mit1C target detection comparison between PerfeCTa master mix and PCRBio master mix with UNG and dUTP, duplicate samples per titer in Rheonix Assay.

Mit1AA gBlock, Copies/15 µL Reaction	PerfeCTa	PCRBio with UNG and dUTP
500	100%	100%
50	100%	100%
5	50%	50%
2.5	NA	50%
1.25	NA	0
0.5	0	0

**Table 6 microorganisms-11-02805-t006:** Additional Mit1C target detection comparison between PerfeCTa master mix and PCRBio master mix with UNG and LoD.

Mit1AA gBlock, Copies/15 µL Reaction	PerfeCTa * No. of Positive Replicates (%)	PCRBio No. of Positive Replicates (%)
6.25	3 (100%)	3 (100%)
3.12	3 (100%)	3 ** (75%)
1.56	3 (100%)	2 ** (50%)
0.78	0 (0%)	1 ** (25%)
0.39	NA	0 (0%)

* Three replicates were run of each concentration. ** Four replicates were run of each concentration.

**Table 7 microorganisms-11-02805-t007:** Testing of DNA samples obtained via manual extraction from fresh produce commodities and one prepared dish unseeded and seeded with different amounts of *C. cayetanensis* oocysts.

Produce	No. Oocysts Seeded	Samples Examined	qPCR Results	Rheonix C. cayetanensis Assay
lettuce	5	1	pos	pos
	10	2	pos	pos
	200	2	pos	pos
	**Total**	**5**		
carrots	0	1	neg	neg
	10	1	pos	pos (after 8 µL DNA used)
	50	1	pos	pos
	200	1	pos	pos
	**Total**	**4**		
pico de gallo	0	1	neg	neg
	5	1	pos	pos (after 8 µL DNA used)
	20	1	pos	pos
	200	1	pos	pos
	**Total**	**4**		
basil	5	1	pos	pos
	10	1	pos	pos
	200	1	pos	pos
	**Total**	**3**		
berries	0	1	neg	neg
	5	1	pos	pos
	50	2	pos	pos
	**Total**	**4**		
cherry tomatoes	0	1	neg	neg
	**Total**	**1**		

**Table 8 microorganisms-11-02805-t008:** Detection of *C. cayetanensis* (rate of positivity) in supernatants of bead-beaten fresh produce samples in the Fully Automated Rheonix C. cayetanensis Assay compared to previous qPCR results based on DNA analysis [4].

Assay: Automated PCR (Rheonix)	Oocysts Numbers Seeded	No. of Samples Analyzed	No. Positive Samples (%)	Previous Results by qPCR Based on [4] (%)
Romaine Lettuce	0	3	0 (0%)	Neg (100)
	100	3	3 (100%)	Pos (100)
	5 *	5	4 (80%)	Pos/Neg (50%)
Cilantro	0	3	0 (0%)	Neg (100)
	100	3	3 (100%)	Pos (100)
	5 *	5	4 (80%)	Pos/Neg (75%)

* Each sample seeded with 5 oocysts (bead-beating wash pellet in 1 mL buffer) was run in quadruplicate. Positive samples had at least one positive replicate, with one sample of romaine lettuce having 2 positive replicates, one sample of cilantro having 2 positive replicates, and two samples of cilantro having three positive replicates.

**Table 9 microorganisms-11-02805-t009:** Oocyst-seeded raspberry sample inhibition results improved in the fully automated assay after dilutions.

Matrix	Original Oocysts per mL	Matrix Dilution	Diluted Oocysts per mL	Oocysts per Reaction	Mit1C Detection
Raspberry	100	1×	100	4	1/3 (33%)
Raspberry	100	0.5×	50	2	7/9 (78%)
Raspberry	100	0.2×	20	0.8	3/3 (100%)
Raspberry	100	0.125×	12.5	0.5	2/2 (100%)
Raspberry	100	0.1×	10	0.4	2/2 (100%)
Raspberry	50	1×	50	2	2/3 (67%)
Raspberry	50	0.4×	20	0.8	1/3 (33%)
Raspberry	40	0.5×	20	0.8	3/3 (100%)
Raspberry	20	0.5×	10	0.4	3/3 (100%)
Raspberry	10	0.5×	5	0.2	2/4 (50%)
Raspberry	10	0.25×	2.5	0.1	0/3 (0%)
Raspberry	10	0.125×	1.25	0.05	3/3 (100%)

## Data Availability

Data are contained within the article.

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
