# Peer review of "Development and Evaluation/Verification of a Fully Automated Test Platform for the Rapid Detection of Cyclospora cayetanensis in Produce Matrices"

_microorganisms, 2023, doi:10.3390/microorganisms11112805_

Round 1
Reviewer 1 Report
Comments and Suggestions for Authors
Please see my suggestions for improvement below:
Line 18-Can remove word "products", fresh produce stands alone
Line 19-Mit1C target assay was already developed, this study describes the integration into and verification of
Line 21-remove "its ease of use" and change to "simplify and reduce hands-on time for…"
Line 23-missing word "developed" between previously and Mit1C, change "to" to "into"
Line 24-is Rheonix a trademark?
Line 28 and Line 105-Revise, "performance against…" should be reworded
Line 43-method uses a single primer pair and probe?
Line 45-mention improved specificity of mitochondrial target region?
Line 55-replace "in" with "using"
Line 57-samples not sample
Line 58-you are not assessing the platform for environmental samples in this study so should refer only to produce
Line 64-effort not efforts
Line 65 and throughout-italicise species names
Line 71-comma should be after termini
Line 69-endpoint not end point
Line 79-specimens refers to organisms, sample is probably more appropriate
Line 123-reword "were prepared with 50g"
Line 171-spell out full species name the first time it is used, i.e. Eimeria maxima
Line 183-primer and probe sets
Line 203-that's a bit of an exaggeration? The user needs to load the samples and start the instrument at minimum?
Line 217-244-Are the buffers provided by the manufacturer or were they in-house developed or from a previous publication?
Table 3a and b could be combined into 1 table
Include some description of how the probe concentrations and hybridization temperatures were selected.
Line 354-correct to "all of the DNA input amounts"
Line 346-Isn't the purpose of the IAC to detect inhibition in a sample? Why did you chose this approach to assessing potential matrix effects rather than testing previously inhibited samples (ie samples that were inhibited using the standard qPCR?)
Table 5-how many samples were tested? Only percentages are shown. Remove shading from table.
Table 6-missing bottom border. Table formatting throughout is not consistent.
The fact that UDG master mixes were adopted implies some issues with carry-over contamination? Were negative reagents-only controls are included in each run?
Table 7-missing negative basil and lettuce samples? Why only un-spiked cherry tomato sample used?
Line 454-do you mean samples seeded with 50 oocysts or 50 lettuce samples, seeded with how many oocysts?
Line 494-497-this greatly reduces the throughput of the automated assay if you need to use 4 cells to test one sample (assuming most naturally contaminated samples are at the fractional level)
Table 8-expand column width to fit text. Change "expected" to previous qPCR result
Line 544-change to and, not and/or
Line 559-what is the cost of the instrument? What is the estimated cost per sample for reagents and consumables for the hands on method compared to the automated method?
Line 590-performance not performances
Comments on the Quality of English Language
Generally good, see suggestions for improvement above.
Author Response
We would like to thank the reviewer for his/her helpful comments that improved the manuscript. Our responses are found following each comment in bold text.
Line 18-Can remove word "products", fresh produce stands alone. Deleted.
Line 19-Mit1C target assay was already developed, this study describes the integration into and verification of. Deleted “development”
Line 21-remove "its ease of use" and change to "simplify and reduce hands-on time for…". Removed.
Line 23-missing word "developed" between previously and Mit1C, change "to" to "into" Added and changed.
Line 24-is Rheonix a trademark? Yes. The first time the assay is mentioned (abstract) the TM designation has been included as “Rheonix C. cayetanensis™ Assay”. To be consistent with trademark usage requirements, the TM just need to be included once in the document. If the reviewer would like us to add it more times, please let us know.
Line 28 and Line 105-Revise, "performance against…" should be reworded. Changed as “including the same performance for both inclusion and exclusion panels”.
Line 43-method uses a single primer pair and probe? Corrected.
Line 45-mention improved specificity of mitochondrial target region? Added “and the improved specificity of the mitochondrial target region”.
Line 55-replace "in" with "using" Replaced.
Line 57-samples not sample. Corrected.
Line 58-you are not assessing the platform for environmental samples in this study so should refer only to produce. Eliminated “and environmental samples”.
Line 64-effort not efforts. Corrected.
Line 65 and throughout-italicise species names. Rheonix assay names do not include italics. The parasite name is italicised when it is not part of the assay name.
Line 71-comma should be after termini. Added comma.
Line 69-endpoint not end point. Corrected.
Line 79-specimens refers to organisms, sample is probably more appropriate. Changed as “samples”
Line 123-reword "were prepared with 50g" Reworded as “which consisted of 50 g”
Line 171-spell out full species name the first time it is used, i.e. Eimeria maxima. Corrected.
Line 183-primer and probe sets. Corrected.
Line 203-that's a bit of an exaggeration? The user needs to load the samples and start the instrument at minimum? Corrected as “with minimal intervention by the user”.
Line 217-244-Are the buffers provided by the manufacturer or were they in-house developed or from a previous publication? The buffers were in-house developed unless they were specified with manufacturer’s name.
Table 3a and b could be combined into 1 table. Tables 3a and 3b have been combined as Table 3.
Include some description of how the probe concentrations and hybridization temperatures were selected. The description on how the probe concentrations and hybridization temperatures were selected has been added to the text.
Line 354-correct to "all of the DNA input amounts" Corrected.
Line 346-Isn't the purpose of the IAC to detect inhibition in a sample? Why did you choose this approach to assessing potential matrix effects rather than testing previously inhibited samples (ie samples that were inhibited using the standard qPCR?). As the reviewer mentions, the IAC was used to detect inhibition in any commodity and sample. We wanted to see if we could maximize the template input amounts in our assay to improve assay sensitivity without causing inhibition. Unlike the established qPCR assay (2 µl of DNA/20 µl reaction), in the Rheonix Assay, we can add up to 6 µl of template into a 15 µl reaction. We did not include any previously inhibited samples, since in our experience using qPCR, this is very uncommon. In the occasions that we had inhibited DNA, we cleaned the DNA using commercial kits.
Table 5-how many samples were tested? Only percentages are shown. Remove shading from table. Duplicate samples were used for each titer dilution. This is included in the legend of Table 5. The shading was removed from the table.
Table 6-missing bottom border. Table formatting throughout is not consistent. Tables have been consistently formatted.
The fact that UDG master mixes were adopted implies some issues with carry-over contamination? Were negative reagents-only controls are included in each run? No, we had not seen any carry-over contamination. Adding UNG to master mix is a typical practice in the molecular diagnostics industry to eliminate carry-over contamination. We did so for precaution. Negative control only reagents were included in each run.
Table 7-missing negative basil and lettuce samples? Why only un-spiked cherry tomato sample used? The samples were randomly selected from manually DNA extracted samples to include several types of commodities and levels of parasite contamination, that is the reason why not all levels are included in some commodities.
Line 454-do you mean samples seeded with 50 oocysts or 50 lettuce samples, seeded with how many oocysts? Romaine lettuce samples seeded with 50 oocysts. This has been corrected in the text.
Line 494-497-this greatly reduces the throughput of the automated assay if you need to use 4 cells to test one sample (assuming most naturally contaminated samples are at the fractional level). We agree that most naturally contaminated samples will be expected to be at fractional level. The whole supernatant, distributed in 4 replicates, was run in the assay to compare with the use of the whole supernatant for DNA extraction for qPCR. However, due to the high rate of positivity observed in Romaine lettuce or cilantro (over 75%), the users might want to run single replicates of those commodities and if not positive, repeat with a second replicate.
Table 8-expand column width to fit text. Change "expected" to previous qPCR result. Columns were expanded and the text corrected.
Line 544-change to and, not and/or. Changed
Line 559-what is the cost of the instrument? What is the estimated cost per sample for reagents and consumables for the hands on method compared to the automated method? We do not have a comparison of the costs of all reagents and equipment used for the equivalent steps in both methods. We do not think that these estimates need to be added to the manuscript.
Line 590-performance not performances. Corrected.
Reviewer 2 Report
Comments and Suggestions for Authors
The work is very interesting and relevant as it seeks a faster and more effective diagnosis for Cyclospora cayetanensis.
Some suggestions:
On page 6, place the title of table 2 next to the table to make it easier to read.
Many tables do not have the bottom line after the data (table 2, table 3a, table 3b, table 6). Please correct.
In some figures, such as figures 4, 5, 6 and 7, the font size is very small, which makes reading difficult. Wouldn't it be possible to fit the size of the tables?
On page 10 there is a line below the description of the figure. To remove.
On page 14 above the table there is an extra line that needs to be removed.
Author Response
Answers to Reviewer 2: The work is very interesting and relevant as it seeks a faster and more effective diagnosis for Cyclospora cayetanensis.
Thank you for your nice words and for your suggestions to improve the manuscript. Our responses are following each comment in bold text.
Some suggestions:
On page 6, place the title of table 2 next to the table to make it easier to read. The title was placed next to the table.
Many tables do not have the bottom line after the data (table 2, table 3a, table 3b, table 6). Please correct. We added the bottom line in all the tables after the data.
In some figures, such as figures 4, 5, 6 and 7, the font size is very small, which makes reading difficult. Wouldn't it be possible to fit the size of the tables? The font size in the figures have been increased.
On page 10 there is a line below the description of the figure. To remove. Removed.
On page 14 above the table there is an extra line that needs to be removed. Removed.
Reviewer 3 Report
Comments and Suggestions for Authors
Key words should be different from the title.
The introduction addresses the problem well.
The methodology is feasible.
The titles of the tables and figures should be self-explanatory, especially in the material and methods and results, where abbreviations should be avoided.
Author Response
Answers to Reviewer 3-
Key words should be different from the title. The keywords have been modified.
The introduction addresses the problem well. Thanks for your comment.
The methodology is feasible. Thanks for your comment.
The titles of the tables and figures should be self-explanatory, especially in the material and methods and results, where abbreviations should be avoided. Thanks for your comment. We avoided abbreviations in the tittles of tables and figures.